# The Distinct Roles of Transcriptional Factor KLF11 in Normal Cell Growth Regulation and Cancer as a Mediator of TGF-β Signaling Pathway

**DOI:** 10.3390/ijms21082928

**Published:** 2020-04-22

**Authors:** Lili Lin, Sven Mahner, Udo Jeschke, Anna Hester

**Affiliations:** 1Department of Obstetrics and Gynecology, University Hospital, LMU Munich, Marchioninistr. 15, 81377 Munich, Germany; drAlee1990@outlook.com (L.L.); sven.mahner@med.uni-muenchen.de (S.M.); anna.hester@med.uni-muenchen.de (A.H.); 2Department of Gynecology and Obstetrics, University Hospital Augsburg, Stenglinstraße 2, 86156 Augsburg, Germany

**Keywords:** KLF11, TGF-β, TIEG, cell cycle, apoptosis, cell growth, cancer

## Abstract

KLF11 (Krüppel-like factor 11) belongs to the family of Sp1/Krüppel-like zinc finger transcription factors that play important roles in a variety of cell types and tissues. KLF11 was initially described as a transforming growth factor-beta (TGF-β) inducible immediate early gene (TIEG). KLF11 promotes the effects of TGF-β on cell growth control by influencing the TGFβ–Smads signaling pathway and regulating the transcription of genes that induce either apoptosis or cell cycle arrest. In carcinogenesis, KLF11 can show diverse effects. Its function as a tumor suppressor gene can be suppressed by phosphorylation of its binding domains via oncogenic pathways. However, KLF 11 can itself also show tumor-promoting effects and seems to have a crucial role in the epithelial–mesenchymal transition process. Here, we review the current knowledge about the function of KLF11 in cell growth regulation. We focus on its transcriptional regulatory function and its influence on the TGF-β signaling pathway. We further discuss its possible role in mediating crosstalk between various signaling pathways in normal cell growth and in carcinogenesis.

## 1. Introduction

The Krüppel protein regulates the body segmentation in the thorax and anterior abdomen of the Drosophila embryo in Drosophila melanogaster [1]. Krüppel-like factors (KLFs) were first discovered in 1993 and named “Krüppel-like” as mammalian homologs of the Drosophila melanogaster gene Krüppel. The first discovered KLF was named EKLF (also known as KLF1) [2]. Since then, diverse homologous genes have emerged. All of them are now called the family of KLFs [3]. Currently, there are 17 known KLFs in mammals. The functions of the family members are, in some cases, overlapping and, in others, widely distinct [4]. KLFs can be divided into three subgroups based on structural and functional features. Subgroups I and II of KLFs consist of KLF3, KLF5, KLF6, KLF7, KLF8, and KLF12 and of KLF1, KLF2, KLF4, and KLF17, respectively. The remaining KLFs (KLF9, KLF10, KLF11, KLF13, KLF14, KLF15, KLF16) belong to subgroup III [5]. 

Structurally, KLFs are composed of a conserved C-terminal region and widely divergent N-terminal regions. The C-terminus consists of three tandem Cys2His2 zinc finger motifs, which act as a DNA-binding domain. The zinc fingers bind GC-rich sequences (designated as Sp1 sites), including GC boxes (GGGGCGGGG), GT/CACC boxes (GGTGTGGGG), and basic transcription elements (with a preference for the 5′-CACCC-3′core motif), with varying affinities. These binding sequences are widely distributed in promoters, enhancers, and locus control regions (LCRs) of housekeeping genes as well as of tissue-specific and viral genes [6,7] that are necessary for cell proliferation, differentiation, apoptosis and morphogenesis [8,9,10,11]. The N-terminus contains the transcriptional regulatory domains that vary significantly. It consists of acidic transactivation domains, Sin3 interacting domains (SID), or C-terminal binding protein (CtBP)-binding repressor domains that can bind co-factors and contribute to the distinct functions of KLFs [5]. The functions of Sin3 and CtBP might compete because the SID and CtBP-binding domains overlap [5]. Only in several KLFs (KLF1, KLF4, KLF8, KLF11), the nuclear localization signal (NLS)—which causes the nuclear localization of KLFs—is located immediately adjacent to/within the zinc finger motifs. [12]. 

The transforming growth factor-beta (TGF-β) inducible immediate early gene (TIEG) encodes a protein that has the ability to inhibit cell growth in cultured osteoblastic and epithelial cell populations [13,14]. In 1998, a novel Sp1-like zinc finger encoding cDNA was identified, and the gene product was named KLF11 [15]. KLF11 shows a high level of homology with TIEG and was named TIEG2, accordingly. Since then, TIEG has been referred to as TIEG1 or KLF10 [15]. Both TIEG proteins can be assigned to subgroup III of the KLFs. TIEG2 shares 91% homology within the C-terminal zinc finger region with TIEG1 and 44% homology within the N-terminus. However, it does not share any significant homology with any other previously identified protein [15]. TIEG1 and TIEG2 share several proline-rich sequences within the N-terminal domain, a property commonly found in other transcription factors as well [2,16,17,18,19,20]. Both TIEG proteins, like all members of subgroup III, share in the N-terminus a conserved repression motif and an α-helical domain highly similar to the Sin3 interaction domain (SID) of the transcriptional repressor Mad1 [1]. This SID-like domain mediates transcriptional repression by interacting with the histone deacetylase corepressor complex mSin3A [21,22,23]. However, the repressional function can be modified: The phosphorylation of different residues in a region adjacent to the KLF11 SID-domain via the EGF–Ras–MEK1–ERK2 signaling pathway [24] and/or the EGFR/AKT–KLF11 (Thr-56 phosphorylation) signaling pathway [25] disrupts the interaction of the SID-like domain with mSin3A and results in a significant loss of the repressional function.

KLF11 plays a vital role in a variety of cell types and tissues. It potentiates TGF-β-mediated anti-proliferative signaling pathways and thereby inhibits epithelial cell growth [15,26,27]. While TIEG2/KLF11 represents the human isoform [15], Tg3/KLF11 [28] is a murine isoform. Based on the observations that KLF11 is significantly induced by TGF-β and, when artificially overexpressed, mimics TGF-β–induced effects in various epithelial cell systems in vivo and in vitro, it has been speculated that KLF11 might function as a TGF-β effector protein that participates in TGF-β signaling pathways [15,29]. Within this review, we will focus on recent advances describing the different roles and mechanisms of KLF11 between normal cell growth and cell growth disorders, which is particularly important for future analyzes regarding the role of KLF11 in carcinogenesis and which may lead to new insights and discoveries for cancer diagnosis or treatment. 

## 2. Genomic Organization and Characteristic Structural Features of KLF11

KLF11 mRNA is ubiquitously expressed in human tissues, with the highest levels found in the pancreas and in skeletal muscle [15]. It is a nuclear protein but is excluded from the nucleolus [15]. The transcriptional regulatory function of KLF11 is mediated by potent repressor activity located within the N-terminal region of the protein [15]. Table 1 gives an overview of the genomic information of KLF11 in Homo sapiens (humans). Chromosomal localization data refers to human genes and has been obtained from the human genome database of the National Center for Biotechnology Information (NCBI) (https://www.ncbi.nlm.nih.gov/gene).

As a member of the KLFs family, the structure of KLF11 is characterized by the highly homologous C-terminal DNA binding domain containing three Cys2His2 zinc finger motifs that bind the GC-rich sequences (the Sp1-like sites) within promoters [1]. The “Sp1 site” dependent transcription can be regulated: the activity, expression, and/or post-translation modification can be altered with cell growth. A basic tetra peptide within the second and third zinc finger of the KLF11 DNA-binding domain functions as an NLS, which is essential for it to be able to act as a site-specific transcription factor [15,30]. The N-terminal region of KLF11 consists of the three repression domains named R1, R2, and R3. They can be tethered to the DNA through a heterologous DNA binding domain (DBD) to mediate repression. The R1 domain, an alpha-helical repression motif (HRM), has been shown to be essential for the interaction with the co-repressor mSin3A, and this interaction is followed by the inhibition of the transcriptional activation of target genes [23]. Therefore, the R1 repression domain is often referred to as the mSin3A interacting domain (SID) and seems to be the most important domain for facilitating KLF11-mediated transcriptional repression [9,15,23]. 

## 3. KLF11, a Context-Dependent Transcriptional Repressor or Activator

KLFs are known to activate or repress gene expression in various organisms [31], subsequently leading to context-dependent effects that might appear contradictory. TIEG1/KLF10 has been shown to behave as a transcriptional repressor [32], and similarly, KLF11 represses promoter activity in vitro and in vivo through the Sp1-like binding sites in the three repression domains [15]. However, the zinc finger of KLF11 containing the DBD alone is able to activate the transcription [30].

KLF11 represses transcription through the recruitment of the mSin3A–histone deacetylase (HDAC) complex via the SID [23]. Sequence analysis revealed that the SID is highly conserved within the TIEG protein family, indicating that the recruitment of the HDAC complexes via binding mSin3A is a general mechanism of transcriptional repression in these proteins [30]. Nevertheless, in HeLa cells, KLF11-mediated transcriptional repression was not mSin3A-dependent, so the R2 and R3 domains might also play an important role as transcriptional repressors [9,30]. However, the inhibition of transcription by R2 and R3 is less well understood than the repression mediated by R1.

Although KLF11 was originally identified as a transcriptional repressor, recent studies have demonstrated that it is also able to activate transcription in various cell types. Binding to the insulin promotor and activating the transcription of the insulin gene, KLF11 plays a vital role in the function of the endocrine pancreas [33]. KLF11 could also interact with the co-activator p300 and activate the pancreatic-duodenal homeobox-1 gene (Pdx-1), which is an important mediator of pancreatic beta-cell activity [34]. 

As KLF11 can behave as a transcriptional activator or as a repressor, it has been suggested that the divergent function depends on the cell type and the promoter context [15,35]. The intramolecular interactions of the *N*-terminal repression domains and the *C*-terminal activator domain, which could be mediated by protein folding, have a counteracting activity in the context of the full-length state [30]. Thus, KLF11 might act as a context-dependent transcriptional repressor or activator by interacting with coactivators or corepressors to orchestrate the transcriptional regulation of target genes.

## 4. KLF11 Contributes to the Regulation of Normal Cell Growth

TIEG proteins can act as growth-inhibiting and/or pro-apoptotic proteins in different cell types [14,36,37,38,39]. Early research reported that KLF11 inhibits cell growth in pancreatic epithelial cells [14,40] in an osteoblastic cell population [13] and in prostate cells through cell cycle regulation [41]. A proposed mechanism is the interaction with the TGF-β signaling pathway. TGF-β plays an important role in the inhibition of cell growth, the regulation of extracellular matrix components, in cell differentiation and migration, as well as in the induction of apoptosis. The intracellular signaling pathway of TGF-β has been well elucidated [42,43,44]. In the TGF-β-signaling pathway, the Smad7 protein acts as a negative feedback loop that suppresses the TGF-β-induced growth inhibition. KLF11 is now able to transcriptionally silence the Smad7 gene and to disrupt the negative feedback loop. Thereby, KLF11 potentiates the TGF-β-signaling and the inhibitory effects of TGF-β on cell growth [45,46]. KLF11 can be induced by several members of the TGF-β superfamily, including all three TGF-β isoforms, by activin, by BMPs, and by the glial-cell-derived neurotrophic factor [47]. Via transcriptional regulation of further genes that induce apoptosis or cell cycle arrest, KLF11 can also inhibit proliferation [15,23,37].

The importance of KLF11 in cell growth regulation was firmly established by a study demonstrating that its overexpression blocks the proliferation of Chinese hamster ovary (CHO) cells [15]. CHO cells transfected with wild-type KLF11 showed decreased proliferation by 60% compared to cells that were transfected with SID-deleted KLF11 or to normal control CHO cells [37]. Similarly, the PANC1 exocrine pancreatic epithelial cell line transfected with KLF11 showed a significant decrease in cell proliferation and an increase in the number of apoptotic cells [37]. In vivo, transgenic expression of KLF11 in the mouse exocrine pancreas resulted in decreased cell proliferation, increased apoptosis, and reduced organ size [37]. KLF11-transgenic mice displayed downregulation of genes encoding oxidative stress scavengers, such as superoxide dismutase 2 (SOD2) and catalase1, which might contribute to an increased susceptibility to oxidative stress and increased cell death. In embryonic stem cells, cell growth inhibition and transcriptional regulation were observed when KLF11 was transiently or stably expressed [48]. In OLI-neu cells, KLF11 was described as a pro-apoptotic downstream mediator of TGF-β [28,30]. 

Members of the Bcl-2 family are important regulators of apoptosis, deciding whether cells live or die. The ratio between anti-apoptotic (e.g., Bcl-X_L_, Bcl-2) and pro-apoptotic (e.g., Bax, Bak) proteins determines the susceptibility of cells to apoptosis [49]. The promoter region of Bcl-X_L_ contains several Sp1 sites serving as binding elements of KLFs [50]. KLF11 bridges the TGF-β-signaling to the repression of anti-apoptotic Bcl-X_L_ expression and thereby increases apoptosis. Overexpression of KLF11 in OLI-neu cells resulted in apoptotic cell death accompanied by decreased levels of the anti-apoptotic Bcl-X_L_ [39]. On the other hand, KLF11 increased leiomyoma cell proliferation and abolished the anti-proliferative effect of the progesterone antagonist RU486 via integrating with progesterone receptor (PR) signaling [51].

However, one other in vivo study showed contrary effects with less importance of KLF11: The KLF11 gene was inactivated in mice by gene knockout technology, and KLF11-/- mice were normal in growth, bred according to the normal Mendelian genetics, and lived as long as KLF11+/+ mice. Hematological analyzes also revealed no abnormalities in KLF11-/- mice [48]. 

In summary, KLF11 is highly induced by TGF-β and, when overexpressed, mimics the TGF-β induced cell cycle arrest in epithelial cells. However, in vivo, the effects of KLF11 are less consistent and show either inhibition of proliferation or normal cell growth (Table 2).

## 5. The Relation of KLF11 to Cancers

Given the described role of KLF11 in growth regulation, it is not surprising that KLF11 has also been implicated in the development of tumors. In carcinogenesis, the TGF-β signaling pathway plays a dual role characterized by tumor suppression at early tumor stages and enhanced tumor progression at the late stages of the disease [43,59,60]. TGF-β mediates tumor suppression via Smad-dependent signaling, which then regulates the transcription of cell-cycle-associated genes like p15 and p21 [44,61]. But Smad7 also exerts a negative feedback loop by binding to the activated TβR-I, blocking it and preventing phosphorylation [62,63]. Disturbances in the Smad-dependent signaling have recently been shown in human cancers (including pancreatic cancer), and are associated with the ability of tumor cells to escape from the TGF-β–induced growth inhibition [64,65]. TGF-β may also signal through Smad-independent signaling cascades (e.g., Rho-like guanosine triphosphatases, p38, mitogen-activated protein kinase [MAPK], phosphatidylinositol-3-kinase or c-Jun-N-terminal kinase) and induce an epithelial-to-mesenchymal transition (EMT)—which is a key process in the formation of cancer metastasis—of tumor cells, leading to enhanced tumor cell migration and invasion [66,67,68,69,70]. 

KLF11, as discussed above, is an early response transcription factor that potentiates the TGF-β induced growth inhibition in normal epithelial cells by terminating the inhibitory Smad7 loop. In pancreatic cancer cells with oncogenic Ras mutations, this function of KLF11 is inhibited by the oncogenic Erk/MAPK: Erk/MAPK phosphorylates KLF11, which leads to the disruption of the KLF11–mSin3a interaction [46]. This inhibits the binding of the KLF11–Smads complex to the TIE element and leads to a reduced TGF-β-induced c-myc repression and to a reduction of the anti-proliferative effects of TGF-β [40]. Another study showsd that KLF11 is inhibited in CHO cells by the epidermal growth factor (EGF)–Ras–MEK1–ERK2 signaling pathway. Like in the Erk/MAPK pathway, phosphorylation of four serine/threonine sites adjacent to the SID leads to the disruption of the SID–mSin3A interaction [24]. In KRAS oncogenic mutant cancer cells, KLF11 inhibits BrdU incorporation, increases apoptosis, and inhibits the KRAS-mediated foci and agar colony formation. In vivo, KLF11 partly inhibits the growth of pancreatic tumor cells of KRAS mutant xenografts, by inducing cell cycle arrest at the S phase via downregulation of cyclin A2 [11]. Therefore, KLF11 might participate in the functional switch of TGF-β from a tumor suppressor to a tumor promoter. Figure 1 gives an overview of the different roles of the KLF11-mediated TGF-β–TGF-receptor–Smad signaling pathway in normal cells and tumors.

In addition to the inhibition of KLF11 by SID phosphorylation via Erk/MAPK, Buttar et al. [25] suggested an additional model of KLF11-mediated tumor suppression and its antagonism by an oncogenic pathway. KLF11 binds to the GC-rich consensus sequences in the promoter region of cPLA2α, the key rate-limiting enzyme of the oncogenic PGE2 cascade. Following binding, KLF11 represses the cPLA2α promoter by recruiting the chromatin-remodeling complex Sin3a-HDAC to the promotor. In this way, KLF11 behaves as a tumor suppressor gene by repression of the cPLA2α–PGE2 pathway. This mechanism was shown in Barrett’s epithelial cells. EGFR-AKT signaling, which is upregulated in a subset of patients during carcinogenesis in Barrett’s esophagus cancer, leads to the phosphorylation of threonine at position 56 in the R1 domain (SID) of KLF11. This phosphorylation inhibits the KLF11 binding, and the repression of the cPLA2α promoter and the tumor-suppressing effects of KLF11 are inhibited. Probably EGFR can use two different intracellular pathways (ERK2 versus AKT) to inactivate KLF11 via phosphorylation. These phosphorylation events expand our biochemical knowledge about KLF11 to the post-translational effects.

On the contrary, direct tumor-promoting effects of KLF11 have also been described. In hepatocellular carcinoma, KLF11 had a significant influence on proliferation and apoptosis. It also promoted local invasion and distant migration by suppressing the Smad7 transcription via binding to the Smad7 promoter or by directly upregulating the Smad2/3 expression [71,72]. Ji Q et al. demonstrated that the relative Twist1 promoter region activity increased gradually with increasing KLF11 levels in the plasma [73]. Therefore, they speculated that KLF11 might regulate gastric cancer migration and invasion by increasing the Twist1 expression, which is essential for EMT [73].

KLF11-methylation-dependent inactivation and downregulation occurs in several malignancies, including leukemia, myeloproliferative disorders, esophageal adenocarcinoma, pancreatic cancer, germ cell tumors, ovarian cancer, and head and neck cancer, supporting its candidacy as an actual tumor suppressor gene in humans [25,37,74,75,76,77,78]. KLF11 is involved in the progression of a wide variety of cancers, such as ovarian cancer and pancreatic cancer [40,78]. In breast cancer, the KLF11 promotor is also hypermethylated, and the hypermethylation is associated with low expression of KLF11. KLF11 hypermethylation might be associated with higher rates of metastases [79]. Similar results were found in uterine fibroids and in myelodysplastic syndrome [74,80], suggesting that DNA methylation to regulate KLF11 expression might be a key event that directly contributes to tumorigenesis.

It was recently reported that the microRNA miR-30d increased the survival of BT474 and MDA-MB-231 breast cancer cells. It was shown that it inhibited apoptosis and increased Bcl-2 expression, while it reduced the Bax protein levels. This influence of miR-30d on breast cancer cell growth, metastasis, and EMT is dependent on a low level of KLF11 and on a high level of pSTAT3. KLF11 is a direct target of miR-30d, and KLF11 and pSTAT3 expression are regulated by miR-30d [81]. 

MiR-30 also reduced the profibrogenic TGF-β signaling in hepatic stellate cells by suppressing the KLF11 expression and thus enhancing the negative feedback loop of TGF-β signaling imposed by Smad7 [82]. LincRNA-p21 reduced the availability of miR-30 [83]. Besides miR-30d, overexpression of miR-10b in hepatocellular carcinoma (HCC) promoted HCC cell migration and invasion. MiR-10b downregulated KLF4, which is the inhibitory transcriptional factor of KLF11. In this way, KLF11 was upregulated, which promoted HCC EMT [72].

All these results suggest that KLF11 plays a crucial role during tumorigenesis and development. (Table 3.).

## 6. Conclusions

In the last decade, the transcription factor KLF11 has come into focus in cancer research. However, the characterization of the functions of KLF11 is still in its infancy. Recent data indicates that KLF11 can affect cell growth and carcinogenesis through multiple mechanisms. KLF11 is not only a basal promoter element involved in constitutive gene transcription but also plays multiple roles in the modulation of transcription. It may respond to a particular signal in one cell type but not in another, respond differently to the same signal, or respond to different signals differently in different cells. It might act as a context-dependent transcription regulator and is linked to growth-related biological processes. 

Moreover, KLF11 might play an important role in the development of tumors. However, the role of KLF11 in normal cell growth regulation and cancer is diverse, and other KLFs might induce relevant transcription responses as well. Therefore, a full understanding of the role of KLF11 in growth control and tumor progression will not only requires an analysis of the individual factor KLF11 but also the identification of all KLFs expressed in the cells of interest and the characterization of the transcriptional context.

## Figures and Tables

**Figure 1 ijms-21-02928-f001:**
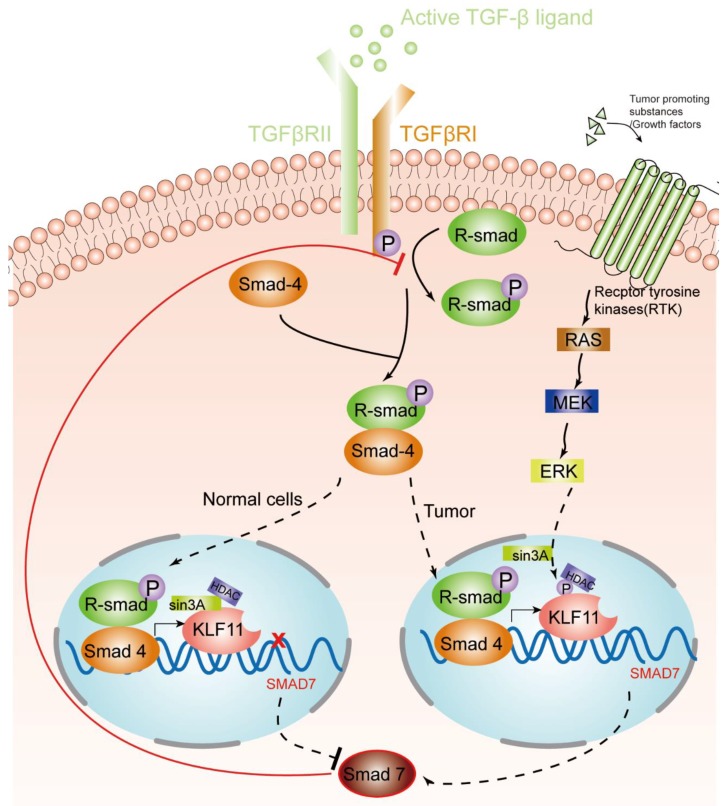
KLF11-mediated modulation of the TGF-β signaling pathway in normal cells and tumors. An activated TGF-β ligand binds to the type 2 domain of the TGF-β receptor, which then recruits and phosphorylates a type 1 receptor. The type 1 receptor then recruits and phosphorylates a receptor-regulated Smad (R-smad). The R-smad then binds to the common smad, Smad 4, and forms a heterodimeric complex. The Smad complex translocates to the nucleus to induce the expression of KLF11. In normal cells, the KLF11-Sin3A-HDAC complex binds to the promoters of Smad7 and represses its expression, which acts as a negative feedback loop of the TGF-β–Smads signaling pathway. However, in some tumors, RAS was activated by tumor-promoting substances/growth factors. Ras phosphates KLF11 by the RAS–MEK–ERK pathway, which leads to the disruption of the KLF11–mSin3a interaction, resulting in the termination of the inhibitory KLF11-mediated Smad7 loop.

**Table 1 ijms-21-02928-t001:** Genomic information and aliases of human Krüppel-like factor 11 (KLF11).

Official Sambo	KLF11
Official Full Name	Krüppel-like factor 11
Gene Type	protein coding
Organism	Homo sapiens
Lineage	Eukaryota; Metazoa; Chordata; Craniata; Vertebrata; Euteleostomi; Mammalia; Eutheria; Euarchontoglires; Primates; Haplorrhini;Catarrhini; Hominidae; Homo
Aliases	FKLF; FKLF1; MODY7; TIEG2; Tieg3
Genomic Localization	Chromosome 2 (2p25.1)
Transcriptional Activity (and Functional Domains)	Activator/Repressor (SID, R2, R3)
Site of Expression	Ubiquitous
Interacting Coactivator and /or Corepressor	mSin3A

TIGE, TGF-β-inducible early gene; FKLF, fetal-beta like globin activating Krüppel-like factor; MODY7, maturity-onset diabetes of the young type 7.

**Table 2 ijms-21-02928-t002:** KLF11 in normal cells/tissues.

Biological Role	Demonstrated Functional Effects	Cell/Tissue	Date	Ref.
KLF11 preserves the structural and functional integrity of the blood-brain barrier	KLF11 activates the promotor of the tight junction proteins occludin and ZO-1	Endothelial cells	2020	[52]
KLF11 induces apoptosis in oligodendroglial cells	KLF11 induces apoptosis by decreasing the levels of the anti-apoptotic protein Bcl-X(L) and inhibiting the transcription of the protein driven by the Bcl-X(L) promoter.	Murine oligodendroglia cells	2007	[39]
KLF11 is a TGF-β-inducible transcription factor with specific domains.	The amino-terminus is essential for the repressive transcriptional effects of KLF11. When the mSin3A domain is lost, the repressive effects are disrupted. The zinc finger containing the DNA-binding domain is essential for the nuclear localization of KLF11 and is able to activate the transcription of reporter genes.	Murine oligodendroglial cells	2007	[30]
KLF11 is required for the browning of human adipocytes caused by PPARγ agonists like rosiglitazone.	KLF11 is induced by PPARγ and increases the mitochondrial oxidative capacity	Human adipocytes	2015	[53]
KLF11 is involved in brown adipocyte differentiation and is highly expressed in brown adipose tissue.	KLF11 induces the expression of the brown adipocyte-specific gene UCP1 by interacting with the UCP1 promotor via GC- and GT-boxes.	Murine mesenchymal stem cells	2010	[54]
KLF11 regulates the estrogen-metabolizing enzyme CYP3A4 in the endometrial epithelium.	KLF11 expression was reduced in the secretory phase endometrium and CYP3A4 was increased. Furthermore, KLF11 colocalized with the corepressor SIN3A/histone deacetylase and repressed the CYP3A4 promoter by deacetylation. This repression was reversed by a KLF11-mutation.	Uterine endometrium	2014	[55]
KLF11 inhibits gluconeogenesis and improves glucose tolerance.	KLF11 inhibits the expression of the gluconeogenic genes phosphoenolpyruvate carboxykinase (cytosolic isoform, PEPCK-C) and peroxisome proliferator-activated receptor γ coactivator-1α (PGC-1α). KLF11-overexpressing mice have less hyperglycemia; KLF11-knockout mice show impaired glucose tolerance.	Mouse hepatocytes, diabetic KLF11-overexpressing and -knockout mice	2014	[56]
KLF11 is involved in regulating the insulin-production of pancreatic beta cells.	KLF11 uses its zinc finger domain and interacts with the coactivator p300 to activate Pdx-1, which is an important mediator of pancreatic beta-cell activity. Maturity onset diabetes of the young (MODY7) variants of KLF11 impair Pdx-1 activation ability.	Human pancreas islet beta cells	2009	[34]
KLF11 inhibits the human proinsulin gene expression.	KLF11 inhibits the proinsulin promotor by binding a GC and a CACCC box	Human pancreatic beta cells	2007	[57]
KLF11 regulates the insulin production in pancreatic beta cells	KLF11 binds to the insulin gene promoter and regulate its activity. Two variants occur in families with early-onset diabetes type 2; in these variants, the transcriptional activity is impaired.	Human pancreatic beta cells	2005	[58]
KLF11 mediates growth inhibition induced by TGF-β cells in pancreatic epithelial cells.	Nuclear KLF11 and Smads3 bind to the core region of the TGF-β inhibitory element of the c-myc promotor and thereby inhibit transcription and cell growth. KLF11 knockdown impairs the growth inhibition.	Human pancreatic epithelial cells	2006	[40]
KLF11 mediates the TGF-β induced growth inhibition by potentiating the Smad-signaling activity.	KLF11 terminates the inhibitory Smad7-loop and therefore potentiates the Smad-signaling. It recruits mSin3a via GC-rich sites to repress the transcription from the Smad7 promoter.	Human pancreatic epithelial cells	2004	[46]

**Table 3 ijms-21-02928-t003:** KLF11 in cancers.

Role in Cancer	Demonstrated Functional Effects	Cancers/Cancer Cell Type	Date	Ref.
KLF11 could be induced in non-small cell lung cancer by radiohyperthermia and might mediate the effects of radiohyperthermia.	KLF11 induced apoptosis and inhibited cell proliferation by elevating intracellular reactive oxygen species. KLF11 knockdown reduced the effects of radiohyperthermia.	Human non-small-cell lung cancer	2019	[84]
KLF11 mediates the tumor-promoting effects of miRNA-30d in breast cancer.	MiRNA-30d increases breast cancer cell survival, inhibits apoptosis, promotes migration and invasion, and mediates the epithelial–mesenchymal transition (EMT) phenotype. MiRNA-30d exerts these effects by targeting KLF11 and activating the STAT3 pathway.	Breast cancer	2018	[81]
KLF11-methylation might be a biomarker for breast cancer diagnosis and prognosis.	The median methylation levels of KLF11 were ≥30% higher than in normal samples. KLF11 methylation might also be associated with a higher risk of metastasis.	Breast cancer	2012	[79]
KLF11 expression is reduced in ovarian cancer.	KLF11 promoter DNA methylation results in downregulated KLF11 expression accompanied by reduced Smad2, Smad3, and Smad7 expression	Human ovarian cancer	2015	[78]
KLF11 is upregulated in gastric cancer an increases gastric cancer cell migration and invasion.	KLF11 increases the Twist-1 expression in gastric cancer cells. The Twist-1 increase is inhibited when KLF11 is silenced.	Human gastric cancer	2019	[73]
KLF11 inhibits prostaglandin E2 (PGE2) synthesis.	KLF11 represses the promotor of the PGE2-synthesizing enzyme cytosolic phospholipase A2∝ by binding and by recruiting the Sin3-histone deacetylase chromatin remodeling complex to the promotor.	Esophageal cancer (Barretts’ esophageal cells)	2010	[25]
KLF11 mediates the-promoting effects of miRNA-10b on EMT development in hepatocellular carcinoma.	MiRNA-10b binds to the 3’UTR and downregulates KLF4, which is an inhibitory transcriptional factor of KLF11. Thereby, KLF11 is upregulated and reduces the expression ofSmad7. This upregulates Smad3, which promotes EMT development.	Human hepatocellular carcinoma	2018	[72]
KLF11 increases the monoamine oxidase (MAO) B expression.	KLF11 increases MAO B at the promotor activity, mRNA, protein, and catalytic activity levels.	Neuroblastoma and liver carcinoma cells	2004	[85]
KLF11 uses the epigenetic regulator heterochromatin protein 1 (HP1) to mediate tumor suppression.	KLF11 recruits HP1 and its histone methyltransferase to promotors of cancer genes to limit the KLF11-mediated gene activation. The impairment of this recruitment impairs tumor suppression.	Pancreatic cancer cells	2012	[86]
KLF11 mediates growth inhibition; the mechanism is disrupted in pancreatic cancer.	In pancreatic cancer cells, the KLF11–Smad3 complex formation is disrupted and the KLF11–Smad3 binding to the TGF-β-inhibitory element of the c-myc-promotor is inhibited. Thereby, the growth inhibitory effect of c-myc-silencing is impaired.	Pancreatic cancer cells	2006	[40]
The KLF11-induced potentiating of the TGF-β-signaling by the termination of the inhibitory Smad7-loop is inhibited in pancreatic cancer.	In pancreatic cancer cells, an Erk/mitogen-activated protein kinase phosphorylates KLF11, which leads to a disruption of the KLF11–mSin3a interaction. The KLF11–mSin3a repression of the Smad7 promotor is reduced, and therefore, Smad7 expression is elevated and Smad7 exerts its negative feedback loop.	Pancreatic cancer cells	2004	[46]
KLF11 is a tumor-suppressor gene inactivated in myelodysplastic syndromes (MDS).	KLF11 is hypermethylated in 15 % of MDS cases, which is associated with a high International Prognostic Scoring System score.	Human myelogenous leukemia cells	2010	[74]

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
