# Peer review of "The Distinct Roles of Transcriptional Factor KLF11 in Normal Cell Growth Regulation and Cancer as a Mediator of TGF-β Signaling Pathway"

_ijms, 2020, doi:10.3390/ijms21082928_

Round 1
Reviewer 1 Report
The authors seem to have produced a reasonably comprehensive account of KLF11 and the TGF-ß signalling pathway. The material is presented in a logical manner. The review cites several reviews. In my opinion a review should cite the primary literature exclusively or at least acknowledge that the statements included are from reviews.
There are many typographical errors, in fact too many to list. All abbreviations should be defined upon first appearance. There are probably journal guidelines for this.
Table 2 is a catalogue of references. I think this would be better placed in supplementary material. The review is focused on cancer so all these different functions are not strictly necessary within the text.
Author Response
Revision 1
The authors seem to have produced a reasonably comprehensive account of KLF11 and the TGF-ß signaling pathway. The material is presented in a logical manner. The review cites several reviews. In my opinion, a review should cite the primary literature exclusively or at least acknowledge that the statements included are from reviews.
Revise: Changed some cited reviews to its primary literature that appoint the point, dig into the depth. Besides of some point that was summarized by some pretty reviews.
There are many typographical errors, in fact too many to list. All abbreviations should be defined upon first appearance. There are probably journal guidelines for this.
Revise: Already add the identification of abbreviations which are not defined first appearance. Mostly in the Abstract. Have already Re-read the guideline of this journal and re-edited the whole article.
Table 2 is a catalogue of references. I think this would be better placed in supplementary material. The review is focused on cancer so all these different functions are not strictly necessary within the text.
Revise: Delete Original Table2.Spit original table 2 to “Table 2 KLF11 in normal cells/tissues” and “Table 3 KLF11 in cancers”. And have deleted the unnecessary mentioned functions.
Reviewer 2 Report
This article focuses on KLF11 relative to the TGF beta signaling pathway. The author wrote the manuscript well and the article is interesting. However, there are some questions remain when the audience reads this paper.
1, It would be better to separate between normal cells or tissue and diseases in Table 2 because it is more apparent how the function of KLF11 works for the progress of diseases.
2, This article discussed KLF11 works as a key role in cell proliferation, differentiation, and apoptosis in several types of cells. It would be better to write the cell differentiation through KLF11 and TGF beta in detail.
3, If it is possible, I would suggest The author makes mention that KLF11 will be applied to cancer treatment in the Conclusion.
Author Response
Revision 2
This article focuses on KLF11 relative to the TGF beta signaling pathway. The author wrote the manuscript well and the article is interesting. However, there are some questions remain when the audience reads this paper.
1, It would be better to separate between normal cells or tissue and diseases in Table 2 because it is more apparent how the function of KLF11 works for the progress of diseases.
Revise: Delete Original Table2.Spit original table 2 to “Table 2 KLF11 in normal cells/tissues” and “Table 3 KLF11 in cancers”. And have deleted the unnecessary mentioned functions.
2, This article discussed KLF11 works as a key role in cell proliferation, differentiation, and apoptosis in several types of cells. It would be better to write the cell differentiation through KLF11 and TGF beta in detail.
Revise: There is no paper particularly talk about this topic, just one paper I keep it the Table 2.” KLF11, but not KLF15, was essential for UCP1 expression during brown adipocyte differentiation of muBM3.1”. It is a common role of other KLFs family members but not KLF11.
3, If it is possible, I would suggest The author makes mention that KLF11 will be applied to cancer treatment in the Conclusion.
Revise: In the third paragraph of conclusion part, I add the sentence “As a result, in this review, we will focus on the latest advances on the regulation and function of KLF11 and highlight their molecular mechanisms, different biological roles in normal cell regulation and cancer to provide a novel therapeutic strategies to cancer prevention and treatment.”
Round 2
Reviewer 1 Report
The authors have made a good effort to update the reference list. There are still many typographical errors.
Author Response
The authors have made a good effort to update the reference list. There are still many typographical Errors.
Answer: Thank you for the Evaluation. We revised the whole manuscript and all typographical Errors, we could identify.